# Implementing Systems Modelling and Molecular Imaging to Predict the Efficacy of BCL-2 Inhibition in Colorectal Cancer Patient-Derived Xenograft Models

**DOI:** 10.3390/cancers12102978

**Published:** 2020-10-14

**Authors:** Alice C. O’Farrell, Monika A. Jarzabek, Andreas U. Lindner, Steven Carberry, Emer Conroy, Ian S. Miller, Kate Connor, Liam Shiels, Eugenia R. Zanella, Federico Lucantoni, Adam Lafferty, Kieron White, Mariangela Meyer Villamandos, Patrick Dicker, William M. Gallagher, Simon A. Keek, Sebastian Sanduleanu, Philippe Lambin, Henry C. Woodruff, Andrea Bertotti, Livio Trusolino, Annette T. Byrne, Jochen H. M. Prehn

**Affiliations:** 1Precision Cancer Medicine Group, Department of Physiology and Medical Physics, Royal College of Surgeons in Ireland, D02 YN77 Dublin, Ireland; aliceofarrell@rcsi.ie (A.C.O.); monika.jarzabek@gmail.com (M.A.J.); ianmiller@rcsi.ie (I.S.M.); kateconnor@rcsi.ie (K.C.); liamshiels@rcsi.ie (L.S.); adamlafferty@rcsi.ie (A.L.); kieronwhite@rcsi.ie (K.W.); annettebyrne@rcsi.ie (A.T.B.); 2Department of Physiology and Medical Physics, Centre for Systems Medicine, Royal College of Surgeons in Ireland, D02 YN77 Dublin, Ireland; andreaslindner@rcsi.ie (A.U.L.); steven.carberry.is@gmail.com (S.C.); lucantoni.federico@hotmail.it (F.L.); mmvillamandos@rcsi.ie (M.M.V.); 3UCD School of Biomolecular and Biomedical Science, UCD Conway Institute, University College Dublin, Belfield, D04 V1W8 Dublin, Ireland; emer.conroy@ucd.ie (E.C.); william.gallagher@ucd.ie (W.M.G.); 4Candiolo Cancer Institute—FPO IRCCS, Candiolo, 10060 Torino, Italy; eugenia.zanella@ircc.it (E.R.Z.); andrea.bertotti@ircc.it (A.B.); livio.trusolino@ircc.it (L.T.); 5Department of Oncology, University of Torino, Candiolo, 10060 Torino, Italy; 6Department of Epidemiology and Public Health Medicine, Royal College of Surgeons in Ireland, D02 YN77 Dublin, Ireland; patdicker@rcsi.ie; 7The D-Lab, Department of Precision Medicine, GROW—School for Oncology, Maastricht University, 6229 ER Maastricht, The Netherlands; s.keek@maastrichtuniversity.nl (S.A.K.); s.sanduleanu@maastrichtuniversity.nl (S.S.); philippe.lambin@maastrichtuniversity.nl (P.L.); h.woodruff@maastrichtuniversity.nl (H.C.W.); 8Department of Radiology and Nuclear Imaging, Maastricht University Medical Center, 6229 HX Maastricht, The Netherlands

**Keywords:** ABT-199, Venetoclax, colorectal cancer, BCL-2, FOLFOX, PDX, preclinical imaging, radiomics, systems biology, deterministic modelling

## Abstract

**Simple Summary:**

Drugs that sensitise tumours to chemotherapy by enhancing cell death signalling are of significant clinical interest. However, it is challenging to determine which colorectal cancer patients may benefit from such sensitisers. The ability to predict this would be advantageous. Here we show that protein profiling combined with mathematical modelling identifies responsive tumours. Using our modelling method, we predicted the effect of adding a sensitizer drug to chemotherapy in two patient-derived colorectal tumours. We grew the tumours in mice, treated animals with these drugs and performed PET/CT imaging. The predicted “sensitive” tumours were smaller when the sensitising drug was added to chemotherapy whilst it did not further reduce tumour size in “non-sensitive” tumours, thus validating our prediction. PET imaging also supported our predictions. CT analysis (radiomics) revealed features that distinguished the two tumours. This was the first application of radiomic analyses to PDX derived CT data.

**Abstract:**

Resistance to chemotherapy often results from dysfunctional apoptosis, however multiple proteins with overlapping functions regulate this pathway. We sought to determine whether an extensively validated, deterministic apoptosis systems model, ‘DR_MOMP’, could be used as a stratification tool for the apoptosis sensitiser and BCL-2 antagonist, ABT-199 in patient-derived xenograft (PDX) models of colorectal cancer (CRC). Through quantitative profiling of BCL-2 family proteins, we identified two PDX models which were predicted by DR_MOMP to be sufficiently sensitive to 5-fluorouracil (5-FU)-based chemotherapy (CRC0344), or less responsive to chemotherapy but sensitised by ABT-199 (CRC0076). Treatment with ABT-199 significantly improved responses of CRC0076 PDXs to 5-FU-based chemotherapy, but showed no sensitisation in CRC0344 PDXs, as predicted from systems modelling. ^18^F-Fluorodeoxyglucose positron emission tomography/computed tomography (^18^F-FDG-PET/CT) scans were performed to investigate possible early biomarkers of response. In CRC0076, a significant post-treatment decrease in mean standard uptake value was indeed evident only in the combination treatment group. Radiomic CT feature analysis of pre-treatment images in CRC0076 and CRC0344 PDXs identified features which could phenotypically discriminate between models, but were not predictive of treatment responses. Collectively our data indicate that systems modelling may identify metastatic (m)CRC patients benefitting from ABT-199, and that ^18^F-FDG-PET could independently support such predictions.

## 1. Introduction

Several chemotherapeutic agents kill cancer cells by initiating apoptosis. Dysfunctional apoptosis is recognised as a key factor in malignant progression and development of chemotherapy resistance in cancer [1,2]. The mitochondrial apoptosis pathway plays an important role in chemotherapy-induced cell death [1]. BCL-2 family proteins are master regulators of mitochondrial apoptosis, and act on mitochondria to induce, or inhibit, mitochondrial outer membrane permeabilization (MOMP). MOMP is achieved through the activation and homo-oligomerisation of two channel-forming, pro-apoptotic multi-domain proteins within the B-cell lymphoma 2 (BCL-2) family, bcl-2-like protein 4 (BAX) and Bcl-2 homologous antagonist/killer (BAK). Monomeric activated BAX and BAK are sequestered and neutralised by a number of anti-apoptotic multi-domain proteins within the BCL-2 family. The anti-apoptotic function of these proteins are antagonised by a third set of proteins within the family, the Bcl-2 homology-3 (BH3)-only proteins. These are activated by transcription or post-translational modification in response to cellular stress [3,4]. Thus, whether cells undergo mitochondrial apoptosis or not is regulated by a complex network of BCL-2 protein interactions.

Small molecule apoptosis sensitisers, such as BCL-2 antagonists restore or induce cell death signalling in cancer cells. There are now several selective BCL-2 antagonists targeting anti-apoptotic BCL-2 family proteins, with the selective BCL-2 antagonist ABT-199 (venetoclax) now approved for the treatment of acute myeloid leukaemia, chronic and small lymphocytic lymphoma patients who received at least one prior therapy [5]. Unlike many haematological cancers, which often overexpress a single anti-apoptotic BCL-2 protein, solid tumours routinely express several pro- and anti-apoptotic proteins [6]. In the setting of solid tumours, it is therefore more challenging to determine which cancer type and which individual patients may best benefit from a given BCL-2 antagonist, highlighting the need for novel systems-based patient stratification tools.

We previously developed and extensively validated a computational tool, DR_MOMP, which determines the sensitivity of cancer cells to undergo apoptosis [7,8,9]. DR_MOMP is a systems model based on ordinary differential equations (ODEs). It incorporates BCL-2 protein levels of five BCL-2 family members BAK, BAX, BCL-2, BCL-2-like 1-long (BCLxL), and myeloid leukaemia cell differentiation protein (MCL-1), measured by quantitative Western blotting or reverse phase protein array analysis, their interaction kinetics, as well as translocation and degradation kinetics of these proteins to calculate the amount of BH3-only proteins (‘stress dose’) required to induce the process of MOMP [7,9]. DR_MOMP hence delivers for each tumour/cell type a numerical value (‘stress dose’) that is a prognostic marker of clinical outcome in colorectal cancer (CRC) [7,8,9], and capable of predicting the response of CRC [7] and breast cancer cells [10] to genotoxic therapy. Here, we extended the application of DR_MOMP to include the effect of ABT-199 on the apoptosis signalling network, and to determine whether the deterministic model could also be used as a stratification tool for the BCL-2 antagonist ABT-199 in in vivo CRC patient derived xenograft (PDX) models (Figure 1). A secondary, hypothesis generating-aspect of the research question was to investigate whether tumour glucose metabolism and/or radiomics features derived from preclinical ^18^F-Flourodeoxyglucose positron emission tomography/computed tomography (18F-FDG-PET/CT) supported model predictions.

## 2. Results

### 2.1. In Vivo PDX Response to ABT-199 Combined with FOLFOX Validates DR_MOMP Predictions

As per our previous studies that demonstrated the ability of DR_MOMP to accurately predict responses of breast cancer cells to BCL-2 antagonists when combined with chemotherapy [10], we started our investigations by confirming that DR_MOMP can predict the sensitisation of CRC cell lines to genotoxic therapeutics by the BCL-2 antagonist ABT-199. We determined BCL-2 protein profiles in DLD-1 and HCT 116 colon cancer cells by quantitative Western Blotting (Appendix A). We next calculated the ‘stress dose’ required to induce mitochondrial apoptosis in response to genotoxic chemotherapy by employing DR_MOMP systems modelling. DR_MOMP calculated an equivalent apoptosis sensitivity in HCT-116 and DLD-1 cells (Appendix A). Upon treatment with 5-FU/OX, as predicted by the model, DLD-1 and HCT-116 cells did not show a significant difference in mean cell survival (*t*-tests, adjusted Holms *p* > 0.49; statistic performed on data presented in Appendix A). However, DR_MOMP analysis indicated that HCT-116 were predicted to be more efficiently sensitised to chemotherapy by ABT-199 than DLD-1 cells. ABT-199 concentration ranges were chosen to reflect therapeutic plasma concentrations previously been shown to reach approximately 5 µM [11]. To validate these model predictions, we performed a 6 × 6 dose matrix assay using a wide range of 5-fluorouracil (5-FU)/oxaliplatin (OX) and ABT-199 concentrations (Appendix A). Data were also processed using isobologram analysis to evaluate drug interactions (Appendix A). Cell survival analysis confirmed the greater synergistic effect in HCT-116 cells at high ABT-199 concentrations, in line with DR_MOMP predictions.

To explore the above confirmed ability of DR_MOMP to predict responses to BCL-2 antagonists in a pre-clinical in vivo setting, we first profiled BCL-2 proteins in several metastatic CRC PDX tissue samples that were all originally derived from metastatic (m)CRC patient liver metastases, established by Bertotti et al. [12,13]. We identified two PDXs, CRC0344 and CRC0076, which differed in their predicted responses to genotoxic chemotherapy, and their predicted sensitisation by ABT-199. BCL-2 family protein levels in PDX samples were quantified using cell lines with known absolute BCL-2 protein levels as standards, and were in line with previous studies in CRC tissue samples [7,8]. Data are presented in Appendix A and Figure 2A. Western blots are provided in Appendix A. Using these BCL-2 protein profiles, we calculated the intrinsic sensitivity of the two models to genotoxic stress caused by 5-FU either with or without ABT-199, varying the ABT-199 doses between 0 to 10 μM (Figure 2B, Appendix A). In the absence of ABT-199, the calculated DR_MOMP stress dose of CRC0076 was 4271 nM, indicating resistance to 5-FU/OX treatment in this model, which is in line with previous findings [7]. CRC0344 showed a lower DR_MOMP stress doses of 2601 nM (Figure 2B) indicating a higher sensitivity to 5-FU/OX treatment. In the presence of a 10.0 µM ABT-199 dose, the model predicted that the DR_MOMP stress dose was reduced in both models. Of note, the model predicted that the stress dose in CRC0076 was reduced to 1506 nM, thus rendering it potentially more vulnerable to 5-FU/OX treatment upon the introduction of ABT-199. BCL-2 concentrations were 5772 nM and 1980 nM for PDX models CRC0076 and CRC0344 respectively (Figure 2A). Overall, PDX CRC0076 tumours were predicted to be more resistant to genotoxic stress than CRC0344, with the addition of 10 µM ABT-199 sensitising this model. Conversely, CRC0344 was sufficiently sensitive to 5-FU/OX alone.

In order to interrogate the effect of FOLFOX, ABT-199, and combination treatment, we performed a blinded in vivo efficacy study using the two mCRC PDX models CRC0076 and CRC0344. These models were chosen based on previously confirmed engraftment rates (>90%) and their differing DR_MOMP-predicted values. Models (*n* = 13–14 mice/group) were treated with ABT-199 either with or without FOLFOX (5-FU + folinic acid + OX) [replicating the clinical regimen]. Tumour growth was measured using calipers over 4 weeks (Figure 2C,D). In CRC0076, following 28 days of treatment, combination therapy significantly decreased average tumour size (*p* = 0.02) compared to vehicle. However, there was no significant difference between FOLFOX alone and vehicle-treated tumours (*p* = 0.08) (Figure 2C). Conversely, in CRC0344 following 28 days of treatment, both FOLFOX alone and combination therapy decreased average tumour size compared to vehicle (*p* = 0.0013 and *p* = 0.0001 respectively). Thus, the addition of ABT-199 conferred no additional benefit over FOLFOX alone in this model (Figure 2D). These data are in line with DR_MOMP predictions where CRC0344 was predicted to require a substantially lower genotoxic stress dose than CRC0076, thus making it more sensitive to treatment with FOLFOX alone (Figure 2B). Conversely, CRC0076 was sensitised by the addition of ABT-199.

### 2.2. ABT-199 Treatment In Vivo Leads to Decrease of p53 Upregulated Modulator of Apoptosis (PUMA) Levels in the CRC0076 PDX Model

On treatment day 28, c.4 h after the final dose of ABT-199, tumours taken from three representative mice from each treatment group were analysed using immunohistochemistry (IHC) and Western blot (This sample size is deemed appropriate for a pharmacodynamic study as previously outlined by Workman et al. [14]). Tissue was used to study the effects of ABT-199 in the two PDX models.

In CRC0076, we found lower levels of p53 upregulated modulator of apoptosis (PUMA) in the three samples treated with ABT-199 alone when compared to vehicle (Figure 3A, *p* = 0.049). Moreover, the combination treated CRC0076 tumours also showed reduced PUMA levels (Figure 3A, *p* = 0.02). BCL2-Interacting Mediator of cell death (BIM) levels also tended to be lowered by ABT-199 treatment, but this effect was more variable than the effect observed with PUMA. In contrast, we did not observe an effect of ABT-199 on PUMA or BIM levels in the FOLFOX sensitive CRC0344 model (Figure 3B). Full Western blots are presented in Appendix A. Histological examination of CRC0076 (ANOVA *p* = 0.53) and CRC0344 PDX tumours (ANOVA *p* = 0.63) revealed no statistically significant change between the treatment groups in cleaved caspase-3 levels after 28 days of treatment (Figure 3C,D). Our data suggest a numerical trend towards decreased cell proliferation (Ki-67 staining) in CRC0076 (which expresses high levels of BCL-2) following ABT-199 treatment (Appendix A).

### 2.3. Decreased Tumour Glucose Uptake as an Early Biomarker of Response to FOLFOX and ABT-199 Combination Therapy

We next sought to determine whether variability in tumour glucose uptake and metabolism was an early response biomarker in the predicted combination responder, CRC0076, and the predicted FOLFOX alone responder, CRC0344. For each model, six mice from each treatment group were randomly selected to undergo ^18^F-FDG-PET/CT.

To determine the effect of treatment on glucose uptake and metabolism (indicated by tumour standard uptake value (SUV) calculations) in the PDX tumours, pre- and post-treatment scans (taken 2 weeks after treatment initiation) were performed in the same animals (representative images shown in Figure 4A). When comparing pre- and post-treatment SUVs, a significant post-treatment decrease in glucose uptake and metabolism was only evident (decreased SUVmean) in the combination group of the CRC0076 model (*p* = 0.01). In the CRC0344 model no significant change in post-treatment glucose uptake and metabolism was observed in any treatment group (Figure 4B).

### 2.4. Radiomic CT Feature Analysis in DR_MOMP Predicted Combination-Only Responder (CRC0076) and FOLFOX Alone Responder (CRC0344) PDX Models

To further mine the available PET/CT imaging data (Section 2.3, Figure 4), we sought to interrogate available CT data using a radiomic pipeline. This hypothesis-generating study aimed to assess whether radiomic features extracted from pre-treatment images could predict end-study (4- weeks) treatment outcome in the combination-only responder (CRC0076) and/or the FOLFOX alone responder (CRC0344).

CT data from pre-treatment and 2 weeks post-treatment mouse tumour scans underwent additional radiomic analysis (workflow and image segmentation described in Figure 5A–C); Firstly, the relationship between CT-derived radiomic features and the underlying phenotype of DR_MOMP predicted combination therapy responder (CRC0076) and FOLFOX alone responder (CRC0344) models was studied. Subsequently, the baseline CT features that were able to discriminate between models in univariate analysis were interrogated in scans taken at 2-weeks post treatment to assess their ability to predict end study outcome. For each PDX (CRC0076 and CRC0344) *n* = 36 pre-treatment images were used as a training cohort, whilst *n* = 9 pre-treatment images were retained as a validation cohort. The training cohort was used both to establish a radiomics pipeline (not previously established for PDX-derived CT data) and to identify potential radiomic biomarker candidates for further analysis.

Ultimately, seven features best discriminated between the DR_MOMP predicted combination therapy response (CRC0076) and the FOLFOX alone responder (CRC0344) (Figure 5D). Using the *n* = 36 training cohort, a radiomic random forest classifier was trained to separate the two PDXs using these seven identified features. The random forest classifier was then tested to determine if the two PDXs in the validation cohort of pre-treatment scans (*n* = 9) could be successfully separated. Performance of the radiomics classifier in the validation dataset was shown to have an area under the curve (AUC) of 0.80 (95% CI 0.408–1, Figure 5E).

We next assessed whether any of the seven CT-derived radiomic features could predict end-study (4-weeks) treatment outcome in the combination-only responder (CRC0076) and/or the FOLFOX alone responder (CRC0344). For all PDX treatment groups (vehicle, ABT-199, FOLFOX, combination in CRC0076 and CRC0344), the value of each feature at 2 weeks was compared to the pre-treatment values to determine treatment effect on the radiomics feature after 14 days. As before, the post-treatment CRC PDX CT images (*n* = 45) were segmented and radiomics features extracted. Data is presented in Appendix A.

In CRC0076 tumours treated with ABT-199, the Wavelet.HHL_ngtdm_Busyness feature intensity was significantly higher (*p* = 0.04526) after 2 weeks of treatment compared to its pre-treatment intensity (Appendix A). Nevertheless, this did not predict significant treatment response in the ABT-199 cohort over 4 weeks (Figure 2C). All other significant changes in radiomic feature intensity observed after 2 weeks of treatment in either PDXs were seen across all cohorts including vehicle (Appendix A). Ultimately, after two weeks of treatment, none of the 7 identified radiomics features predicted treatment efficacy after 4 weeks in either the combination sensitive CRC0076 tumour or the FOLFOX sensitive CRC0344.

## 3. Discussion

In this study we have investigated whether the BCL-2 systems model DR_MOMP could be exploited as a predictive tool for response to ABT-199 treatment in mCRC patients. In line with previous studies [9,16], the model successfully identified tumour cells resistant to 5-FU/OX treatment. To note, in a previous study using a panel of colon cancer cell lines (*n* = 6) we showed a correlation between DR_MOMP scores and sensitivity to genotoxic drugs 5-FU/OX ([7], Figure 3). Here, we demonstrate that DR_MOMP can identify mCRC tumours that benefit from the addition of ABT-199 to 5-FU/OX-based genotoxic therapy. ABT-199 was approved by the Food and Drug Administration (FDA) in 2016 for treatment of chronic lymphocytic leukaemia and is currently incorporated in more than 150 ongoing clinical trials [17]. However, there are limited trials in solid tumours and none specifically related to CRC, which is surprising considering the well-known relationship between BCL-2 family protein expression to both prognosis and resistance to treatment in CRC [7,17,18,19,20]. CRC tumours show high heterogeneity in their expression of multiple pro- and anti-apoptotic BCL-2 family proteins [7,9]. Tools such as DR_MOMP may, as demonstrated here, identify individual mCRC patients who might benefit from BCL-2 antagonist therapy in future clinical studies.

In vitro, DR_MOMP predicted for DLD-1 and HCT-116 CRC cell lines similar responses to ABT-199 and FOLFOX despite differences in their BCL-2 and BCL-xL levels. In both DLD-1 and HCT-116 cells, the combination of ABT-199 with low concentrations of genotoxic drugs (0.2 µM OX plus 1 µM 5-FU) did not result in more cell death when compared to cultures treated with ABT-199 only (Appendix A), and combination of ABT-199 with higher concentrations of OX plus 5FU (5.0 µM and 25 µM) were required to induce higher levels of cell death. Both cell lines showed a numerically homologous response in their in vitro decrease in cell viability and in silico decrease in DR_MOMP score (Appendix A), however the possibility also remains that low concentrations of OX plus 5-FU were protecting cells from ABT-199 induced death by activating protective stress responses induced by DNA damage.

In the PDX models, which also showed varying BCL-2 family protein levels, while DR_MOMP predicted a differential response to ABT-199 and FOLFOX for CRC0076 and CRC0344, an n-fold greater BCL-2 concentration in one tumour did not translate to an n-fold higher requirement for BCL-2 antagonists (Figure 2A,B). Previous observations have suggested that it may be sufficient to quantify BCL-2 protein levels as stratification tool for ABT-199 [21]. However, our findings in both cell and tumour models, reinforce the assertion that individual BCL-2 family protein levels alone may be insufficient as a sole predictive biomarkers for ABT-199 response in CRC patients. This highlights the power of systems approaches to estimate drug responsiveness and further supports the role of systems-informed biomarkers in this setting.

To validate DR_MOMP predictions in an in vivo, clinically relevant setting, two PDX models were studied. CRC0076 was identified by DR_MOMP as having higher resistance to 5-FU/OX (thus rendering it potentially more susceptible to sensitisation by ABT-199) compared to CRC0344 (Figure 2B). It is noteworthy that CRC0076 is faster growing than CRC0344, reflecting clinical differences in tumour growth rates as evidenced in patients. As predicted, a significant in vivo response to FOLFOX alone was elicited in PDX CRC0344, which was not significantly different to the response to ABT-199 + FOLFOX combination. In contrast, within PDX CRC0076 a significant response was only evident upon treatment with the ABT-199 + FOLFOX combination. The DR_MOMP prediction for which PDX would benefit from the addition of ABT-199 to FOLFOX was thus validated in our in vivo model. As ABT-199 is not an approved drug for the treatment of CRC (and therefore was not used to treat the two patients from which PDXs were derived), we were unable to establish if DR_MOMP was also predictive of patient outcome. Nevertheless, we have previously shown that DR_MOMP can predict response to adjuvant chemotherapy and neoadjuvant chemoradiation therapy in colon and rectal cancer patients. Moreover, DR_MOMP can also be used to calculate a predicted response of patients to BCL-2 antagonists [7,8,9]. Future studies to further determine the translational relevance of DR_MOMP are now warranted (e.g., large-scale PDX co-clinical trials). Ultimately, future implementation of this technology in the clinic will require a tailored clinical trial.

In growing tumours, the pro-apoptotic BH3-only protein PUMA is normally sequestered by BCL-2 and stabilised. In PDX CRC0076, which required ABT-199 for sensitisation, we observed significantly reduced levels of PUMA in ABT-199 treated animals, suggesting that it was released from BCL-2 (which is particularly high in this model) and degraded through the proteasome after ABT-199 treatment [22]. These experiments therefore suggest efficient target engagement in vivo. BIM protein levels are also subject to rapid proteasomal degradation [23], but were not significantly altered by ABT-199, although there was variability between the different groups. In PDX CRC0344, PUMA protein levels were not significantly altered by ABT-199 treatment, suggesting that any PUMA induced is functional (but rapidly degraded) and not sequestered by BCL-2. BIM levels were likewise not altered by the treatments.

Although histological examination of CRC0076 and CRC0344 PDX tumours revealed no significant change between the treatment groups in cleaved caspase-3 levels after 28 days of treatment, it is important to note that cleaved, active caspase-3 is likely only detectable in tissues for minutes to hours [24]. Moreover, BCL-2 family mediated mitochondrial membrane permeabilisation controls both caspase-dependent and caspase-independent cell death pathways such as necroptosis, autophagic cell death and necrosis (reviewed in [25]). Ki-67 staining showed a trend towards reduced cell proliferation in the high BCL-2 expressing CRC0076 PDXs upon ABT-199 treatment. These data suggest that BCL-2 might also control or correlate with cell proliferation, and that ABT-199 inhibits cell proliferation when administrated as a monotherapy (as also seen in ER-positive breast cancer [26]). This effect may be linked to the ability of BCL-2 to improve bioenergetics in cancer cells, which in turn would be inhibited by Bcl-2 and Bcl-xL antagonists [10]. However, we did not observe an inhibitory effect on cell proliferation in the combined treatment group. Here, the induction of apoptosis may have activated a compensatory effect on cell proliferation, a process that is known to occur both physiologically and in the cancer setting (reviewed in [27]).

We next implemented an ^18^F-FDG-PET imaging approach to determine if early changes in tumour glucose uptake and metabolism could predict treatment outcome. The rationale here was that in addition to controlling the process of MOMP during apoptosis, BCL-2 family proteins also regulate the bioenergetics status of cells and may act directly on the mitochondrial respiratory chain as discussed above [28,29,30,31,32]. Thus, we hypothesised that ^18^F-FDG-PET may predict response to therapy at an earlier time point than that seen using other clinical parameters (e.g., change in tumour size). Interestingly after two weeks of treatment, only CRC0076 tumours exposed to combination therapy showed a significant decrease in SUV mean (indicative of overall glucose metabolism). SUVmax (generated using the most intense 4 voxels per volume of interest (VOI) was also investigated but provided no additional insight). In this model, tumours in the combination group were significantly smaller than those treated with FOLFOX or ABT-199 alone. These data suggest that in tumours characterised by high BCL-2 levels, and predicted by DR_MOMP to respond favourably to the addition of ABT-199 to FOLFOX, ^18^F-FDG-PET may be an early response biomarker. Corroborating these results, a recent phase 1b study of ABT-199 combined with tamoxifen in ER and BCL-2–positive metastatic breast cancer treatment responses were pre-empted by metabolic responses using ^18^F-FDG-PET [26]. Importantly, this was the first clinical study to evaluate ABT-199 in a solid tumour. It remains unclear as to why glucose metabolism was not reduced in the CRC0344 model despite both FOLFOX alone and the combination tumours being significantly smaller than untreated controls. One possible explanation is that non-apoptotic functions of BCL-2 family proteins include effects on glucose metabolism [33,34]. In Figure 2A CRC0076 is seen to have elevated BCL-2 but lower BCL-xL levels compared to CRC0344. Moreover, the effect on PUMA and BIM following treatment was variable between the two models (Figure 3). It is possible that differential expression of other BCL-2 family members could underpin variance in glucose metabolism across the two models. Further mechanistic studies are warranted.

Next, CT data from pre-treatment and 2 weeks post-treatment PDX mouse tumour scans from DR_MOMP predicted combination-only responder (CRC0076) and FOLFOX alone responder (CRC0344) PDX models underwent additional radiomic analysis. Radiomics is a multistep process in which radiographic features based on shape, pixel intensities, and texture are extracted both from clinical and pre-clinical radiological images [35,36,37]. Radiomic analysis of CT data was undertaken due to the clinical relevance of CT in the CRC patient setting. In addition to providing novel methods of tumour classification and staging, radiomic analyses may also reveal critical information underpinning the biology of tumours [38]. Here, our CT radiomics pipeline identified 7 radiographic features, extracted from pre-treatment PDX CT images, that when combined into a random forest model were able to classify PDX CRC0076 from PDX CRC0344 (Figure 5D, AUC 0.8 95% CI 0.408–1). These hypothesis-generating data suggest that the model can discriminate the two PDXs based on radiomics features. Interestingly, 6 of 7 of these features were wavelet or multiscale textural features. Such features have previously been employed in patient tumour classification. Specifically, Chaddad et al. used wavelets to classify and discriminate whole slide colon cancer images into different grades of CRC with 93.33% accuracy [39]. This suggests that a CRC PDX CT radiomics pipeline is likely to have useful translational potential. Further validation to understand the predictive potential of wavelet or multiscale textural CT radiomic features in interventional CRC PDX population [40] and clinical trial datasets are warranted.

We subsequently investigated if any of the identified 7 features could individually predict end-study treatment outcome in the DR_MOMP predicted combination-only (CRC0076) responder or FOLFOX alone responder (CRC0344) models. Due to the limitation of treatment group size (*n* = 5–6), modelling was not undertaken on the post-treatment scan feature sets. Rather, we compared the values of each radiomics feature, extracted from 2 week post-treatment images, to pre-treatment values (Appendix A). In CRC0076 PDX tumours treated with ABT-199, after 2 weeks of treatment the intensity of feature Wavelet.HHL_ngtdm_Busyness was significantly higher (*p* = 0.04526) than at baseline (Appendix A). This observation suggests that treatment with ABT-199 alone alters the radiographic phenotype of the tumour. However this feature change was not represented by a change in tumour volume at this time point. Moreover, this change did not precede a significant treatment response after 4 weeks (Figure 2C). All other significant changes in radiomic feature intensity observed after 2 weeks of treatment in either PDX were seen across all cohorts including vehicle (Appendix A). As feature intensities changed equivalently across cohorts, these changes are likely associated with inherent phenotypic changes in PDX tumours over time, including tumour volume. Future studies should consider the implication of tumour size; Specifically, comparing tumours of a similar volume would decrease “noise” and improve comparisons.

A limitation of the current radiomics study is the relatively small number of available PET/CT images compared with previous reports [41]. Here, radiomics was performed using limited available data from a preclinical study powered in the first-instance for drug efficacy comparisons. Moreover, test-retest analyses were not performed. Nevertheless, to the best of our knowledge we have for the first time established a CT radiomics pipeline for application in CRC PDX studies. Such analyses will likely become increasingly important as the PDX field evolves, with an expanding focus on the use of orthotopic models requiring clinically relevant imaging [40,42,43]. Indeed, the translational relevance of pre-clinical radiomic ‘discovery’ studies in animal models is now becoming evident. For example, Zinn et al. recently employed radiomic analyses on glioblastoma magnetic resonance data sets from mice and humans, and showed that selected imaging features are conserved across species [44]. In the current context, future work employing CT scan data from large scale population-based CRC PDX trials is now warranted. Increasing the number of CT images available for analysis could identify more robust predictive radiomics signatures having utility as predictive biomarkers.

## 4. Materials and Methods

### 4.1. Quantitative Western Blotting

Western blot analysis was performed as previously described [8]. Briefly, tumour specimen pellets (tumours were halved following removal and all material was used for blotting) were directly homogenised and mixed with lysis buffer containing 0.5 mmol/L Tris(hydroxymethyl)aminomethane—hydrogen chloride (Tris-HCl) (pH 6.8), 10% glycerin (*w/v*), 2% sodium dodecyl sulfate (SDS) (*w/v*), and protease and phosphatase inhibitor cocktails (Sigma Aldrich, Gillingham, UK). Cell pellets were directly mixed with the lysis buffer. Proteins were transferred onto nitrocellulose membranes using semi-dry transfer. Primary antibodies for BAX, BCL-xL (Cell Signaling Technology, London, UK), BIM (Abcam, Cambridge, UK), BCL-2, MCL-1, BAK, and PUMA (Santa Cruz, Heidelberg, Germany) were used. A 1:5000 dilution was used for β-actin (Sigma Aldrich) and glyceraldehyde 3-phosphate dehydrogenase (GAPDH) (Abcam) proteins as loadings control. After overnight incubation, peroxidase-conjugated secondary antibodies (Chemicon, France or Jackson ImmunoResearch, Ely, UK) at a 1:5000 dilution were added. Images were captured using a LAS-3000 Imager equipped with a cooled 12-bit digital charge-coupled device camera (FUJIFILM UK Ltd. Systems, Sheffield, UK). For BCL-2 family protein quantification HeLa cell extracts (20 µg) were blotted alongside samples for standardisation and quantification. For densitometry analysis, the intensity of protein bands within tumour samples were normalised to the loading control after background subtraction using ImageJ software (National Institutes of Health, Bethesda, MD, USA). Absolute protein levels were calculated by normalising values to HeLa cells. Absolute protein concentrations in HeLa cells were determined with recombinant proteins previously [7].

### 4.2. DR_MOMP Calculation

Cells’ sensitivity to undergo apoptosis was calculated using the ordinary differential equation-based systems model DR_MOMP [7]. Absolute protein levels of BAK, BAX, BCL-2, BCL-xL, and MCL-1 were used to calculate the genotoxic stress dose that induces MOMP (‘stress dose’). ABT-199’s binding affinities were modelled with dissociation constants 0.01 nM for BCL-2, 48.00 nM for BCL-xL and 444.00 nM for MCL-1 [10,21]. Sensitivity of CRC cell lines and PDX models to undergo apoptosis was determined without and with different doses of ABT-199 ranging from 0 to 10 μM. The ABT-199 dose was applied at the same time and for the same duration as the modelled genotoxic stress (12 h). DR_MOMP was executed with Matlab R2007b (V.7.5.0.342; The MathWorks, Natick, MA, USA). Where p53 status was known (DLD-1 cells), DR_MOMP calculations accounted for deficient Puma and Noxa induction [7].

### 4.3. Chemicals

ABT-199 was kindly provided by AbbVie (North Chicago, IL, USA). Oxaliplatin (OX) and Fluorouracil (5-FU) were obtained from Sigma-Aldrich (St. Louis, MO, USA) when used for in vitro studies. OX (TEVA UK Ltd., Castleford, UK), 5-FU (ITH PHARMA, London, UK) and folinic acid (TEVA UK Ltd., Castleford, UK) were kindly provided by Beaumont Hospital Dublin when used for in vivo studies (FOLFOX treatment regimen).

### 4.4. Cell Lines

Authenticated human CRC (HCT-116 and DLD-1) cell lines were provided by the School of Medicine, Dentistry and Biomedical Sciences, Queen’s University Belfast, United Kingdom. Furthermore, all cell lines were Mycoplasma tested regularly. Cells were cultured in RPMI 1640 medium (Sigma-Aldrich). All cell lines were maintained at 37 °C in a humidified atmosphere supplied with 5% CO_2_.

### 4.5. In Vitro Toxicity Assays and Synergy Calculations

Acid phosphatase assay was used to measure cell viability based on the conversion of 4-Nitrophenyl phosphate (pNPP) to p-nitrophenol by cytosolic acid phosphatase [45]. pNPP disodium salt hexahydrate was purchased from Sigma-Aldrich. Cells were grown in a 96 well plate at a density of 1.5 × 10^4^ cells per well and treated with increasing concentrations of 5-FU/OX (with a concentration ratio of 1:5 between OX and 5-FU) in combination with increasing concentrations of ABT-199. After 48 h treatment medium was removed and each well was washed once with 200 μL of 1× phosphate buffered saline (PBS). To each well, 100 μL of assay buffer (0.1 M sodium acetate at pH 5.0, 0.1% Triton X-100, and 7.25 mM p-nitrophenyl phosphate) was added. The plates were then incubated at 37 °C for 2 h. The reaction was finally stopped with the addition of 50 μL and colour development was assayed at 405 nm using a Clariostar plate reader (BMG Labtech, Offenburg, Germany), and applying a path length correction of 200 µL. The non-enzymatic hydrolysis of the pNPP substrate was also determined by including wells with the assay buffer and without any cells. An excel template was used to calculate the fraction affected from viability percentages and the results were analysed with the web version of Chalice Analyzer (Horizon Discovery, Waterbeach, UK) to calculate isobolograms.

### 4.6. Animals

Four- to six-week old (c. 20–25 g) female non-obese diabetic/severe combined immunodeficient (NOD/SCID) mice were purchased from Charles River Laboratories (Cambridge, UK). All animal procedures were approved by the Health Products Regulatory Authority (HPRA) (#AE18982-P099) and the University College Dublin Animal Research Ethics Committee (AREC) (#AREC-16-11).

### 4.7. PDX Efficacy Study

PDX CRC0076 and CRC0344 were selected for in vivo studies based on model predictions and known tumorigenicity of these models in vivo. The researcher performing the in vivo studies was blinded to the predicted DR_MOMP predictions. PDX implantation and expansion was performed as previously described [46].

Mice with established tumours (*n* = 13–14/group/PDX) average volume 360 mm^3^) were assigned into groups ensuring that an even distribution of tumour sizes was represented in each group and that group averages were equal. Groups were subsequently treated for 4 weeks with either (i) ABT-199 (oral gavage, once daily at 100 mg/kg, dissolved in 60% phosal, 50 propylene glycol (PG), 30% polyethylene glycol (PEG) 400 and 10% ethanol), (ii) FOLFOX (all constituent drugs delivered IP, once weekly (on day 3 of each cycle), 5-FU (40 mg/kg in PBS) + Folinic Acid (13.4 mg/kg in PBS) followed two hours later by OX (2.4 mg/kg in 5% glucose/water (*v/v*)), (iii) a combination of both regimens (ABT-199 + FOLFOX), or (iv) all diluent vehicle solutions, as described. Tumour size was evaluated three times per week by caliper and volume calculated using the formula Tumour Volume = 4/3π × (d/2)^2^ × D/2, where d is the minor tumour axis and D is the major tumour axis.

### 4.8. Immunohistochemistry (IHC)

Subcutaneous tumours were excised, rinsed twice in Dulbecco’s-(D)PBS and fixed in 4% formaldehyde for 48 h. Tissues were embedded in paraffin and 5 μm thick sections were cut. Routine hematoxylin and eosin (H&E) staining was performed to facilitate histological evaluation. Standard immunohistochemistry methods were used to detect cleaved caspase-3 on 3 tumours from each treatment group. Briefly, antigen retrieval was performed using pH 6.8 sodium citrate buffer. Primary antibodies (Rabbit polyclonal antibody to cleaved caspase-3, 1:200 dilution or Mouse IgG antibody to Ki-67, 1:150 dilution [both Cell Signaling Technology, London, UK]) were incubated for 1 hr at room temperature, and staining completed using the Lab Vision UltraVision Large Volume Detection System (Fisher Scientific, Dublin, Ireland). All sections were scored blindly and semi-quantitatively by two independent observers. The Allred Score Method based on proportion of positive cells (scored on a scale of 0–5) and staining intensity (scored on a scale of 0–3) was used for cleaved caspase-3. The proportion and intensity were then summed and averaged to produce a total score [15]. For Ki-67 analysis, slides were imaged and colour deconvolution was performed using ImageJ. The Ki-67 proliferation index was determined by calculating Ki-67 stained cells as a percentage of total cells.

### 4.9. ^18^F-FDG-PET/CT Study

Mice were fasted for 6 to 12 h prior to imaging and water provided ad libitum. ^18^F-FDGPET/CT scans was performed on a LabPET4 Triumph scanner (Trifoil Imaging, Chatsworth, CA, USA). PET system calibration was carried out by scanning a phantom scan of known activity as measured in a dose calibrator (Capintec, Florham, NJ, USA). A baseline scan was performed prior to treatment starting and a second scan was performed after 2 weeks of treatment. For imaging, mice were placed in a warming box for 20 min followed by intravenous (IV) tail vein injection under isoflurane anaesthesia, of 8.9 megabecquerels (MBq) (+/− 0.8 MBq) ^18^F-FDG diluted with saline to a volume of 200 µL. Following injection of ^18^F-FDG, mice remained under anaesthesia for a 1-h uptake time, with appropriate heating provided to maintain body temperature. A CT scan was taken immediately prior to a 15 min PET scan. Cone beam CT images (focal spot size, 33 mm) were acquired over ~5 min with 512 projections at a geometric magnification equal to 2. X-ray tube voltage and current were 50 peak kilovoltage (kVp) and 350 µA respectively and exposure was 232 ms. CT images were reconstructed with Cobra (Exxim Computing version 7.3.4, Pleasanton, CA, USA)) to a voxel size of 0.1 × 0.1 × 0.1 mm^3^. PET images were re-constructed with a 3-dimensional (3D) MLEM (Maximum Likelihood Expectation Maximisation) reconstruction algorithm (15 iterations, voxel size of 0.5 × 0.5 × 0.6 m^3^) using the LabPET software (version 1.14.0, Trifoil Imaging) PMOD (Version 3.208, PMOD Technologies, Zürich, Switzerland) software was used to draw VOIs around the entire tumour and SUV was calculated using the following equation: SUV = A/(ID/BW), where A is the activity in the tumour per unit volume, ID/BW is the injected dose per bodyweight and all activities are decay corrected to injection time. SUVmean is the mean SUV value of all voxels included in the VOI and is a measure of glucose uptake and metabolism (i.e., overall glucose metabolism) within the VOI, in this case in the PDX tumours. SUVmax is the average value of the four most intense voxels from each VOI. PET images were co-registered and resliced to CT data for illustration purposes or for guidance as required when delineating CT tumour volumes.

### 4.10. Radiomic Analysis of CRC PDX CT Images

Following reconstruction of image volumes, 45 tumours from *n* = 2 PDX models (CRC0334 and CRC0076) were delineated by three independent researchers. ROI (region of interest) were generated and segmentation carried out on CT scans of PDX tumours using a semi-automatic method using ITK-Snap [47] and PMOD software (Version 3.208, PMOD Technologies). Radiomic feature extraction was performed with PyRadiomics V2.7.7 (Harvard Medical School, Boston, MA, USA) [48]. A total of 833 features were extracted from each pre-treatment CRC PDX CT image, including first order, shape, Gray Level Size Zone Matrix (GLSZM), and Gray Level Run Length Matrix (GLRLM). The feature data set of the 45 pre-treatment CRC PDX images was next split into two cohorts, training and validation, in a 80:20 ratio (*n* = 36 and *n* = 9 respectively, ensuring a 50:50 spilt of the two PDXs (CRC0076 and CRC0344) in each cohort). Feature selection and model training were performed only on the training cohort to prevent information leakage. Features were selected in a three-step approach; All features related to shape (in PDX subcutaneous models, shape features are highly conserved) were removed from further analysis. Inter-correlated features (determined in the training dataset by pairwise spearman correlation (>0.85) and removing the feature with the highest correlation to the remaining features) were likewise removed, yielding 71 features. To further reduce feature dimensionality for this small training dataset a decision tree-based recursive feature elimination method was applied (Figure 5C) and recursive feature elimination was performed using the recursive feature elimination (RFE) function (caret package) with 3 repeats and a resampling method of repeated CV [49]. The performance of the selected number of features was evaluated on the validation dataset via AUC for the ROC. All statistical analysis, model building, and model evaluation were performed in R (version 3.5.2; http://www.r-project.org). The R code used to develop the radiomic classifier to separate PDX CRC0076 and PDX CRC0344 can be found in Appendix A.

### 4.11. Statistical Analyses

Analyses were performed using R (Version 3.4.4, The R Foundation, Vienna, Austria) unless otherwise stated. One-way analysis of variance (ANOVA) with Tukey’s ‘Honest Significant Difference’ (Tukey-HSD) post-hoc tests were used to prove for differences in mean values in vitro and for PUMA and BIM protein level, cleaved caspase-3 and Ki-67 IHC data. *p*-values were not adjusted for multiple comparison among multiple ANOVAs. For in vivo PET analysis, multiple paired t-tests were employed to compare raw PET values for each animal to its own pre-treatment measurement. *p*-values were adjusted for multiple comparison using the Holm method. SAS Software Version 9.3 (SAS Institute Inc., Cary, NC, USA) was employed for in vivo data management and its statistics, including linear mixed models to model CRC0076 and CRC0344 PDX tumour growth separately. Mouse identifier was included as a random effect and time effects were included as a continuous covariate, in addition to treatment, as a factor. Model diagnostics included evaluation of leverage, studentised residuals and best linear unbiased predictions of mouse effects. All pairwise treatment comparisons were performed and corrected for multiple testing (Bonferroni correction). The nominal 5% level of significance was used and therefore each of the 6 pairwise comparisons were evaluated at the 0.8% level. Throughout, *p* values ≤ 0.05 were considered statistically significant. For CT radiomic feature analysis, Wilcoxon rank sum test was applied to compare radiomic features between treatment groups. All radiomic feature statistical analysis was performed using R.

## 5. Conclusions

We demonstrate that the system model DR_MOMP may have potential to effectively predict CRC patients for whom the addition of ABT-199 to a FOLFOX regimen would be advantageous. Tumour SUVmean derived from ^18^F-FDG-PET imaging may represent an early response biomarker. Finally, to the best of our knowledge we have for the first time established a CT radiomics pipeline for application in CRC PDX studies. Such analyses will likely become increasingly important as the PDX field evolves, with an expanding translational focus on the use of orthotopic models supported by clinically relevant imaging. Our findings warrant further validation; firstly in an expanded PDX “population” cohort [12] employing a wider range of DR_MOMP stress-dose predictions. These analyses should be followed by prediction testing in an appropriate clinical setting.

## Figures and Tables

**Figure 1 cancers-12-02978-f001:**
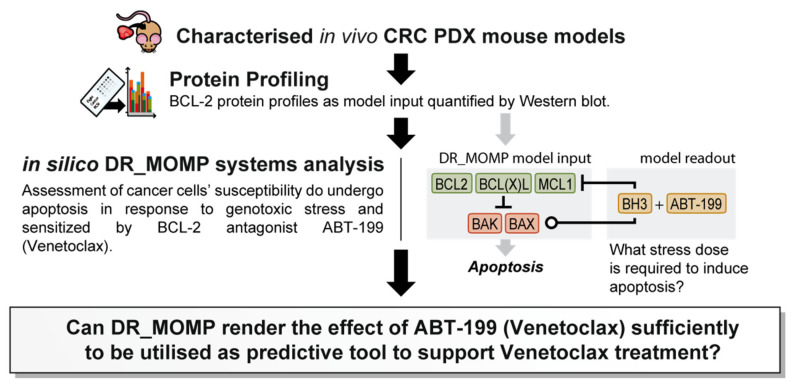
Schematic representation of the experimental workflow. The sensitivity of tumour cells to undergo apoptosis was calculated after quantitative protein profiling using the deterministic systems model DR_MOMP [7]. First, protein levels were calculated by quantitative Western blotting in proteins extracted from colorectal cancer (CRC) patient-derived xenograft (PDX) tumours, using cell lines as standards in which absolute BCL-2 protein levels were previously determined. Protein profiles then served as input into the deterministic systems model, DR_MOMP. The model calculates a ‘stress dose’ (pro-apoptotic ‘BH3 only protein dose’) required to induce mitochondrial outer membrane permeabilization (MOMP), the process leading to mitochondrial apoptosis. Kinetics of the BCL-2 antagonist ABT-199 were included in the DR_MOMP model [10] to determine sensitisation of the apoptosis signalling network by the BCL-2 antagonist.

**Figure 2 cancers-12-02978-f002:**
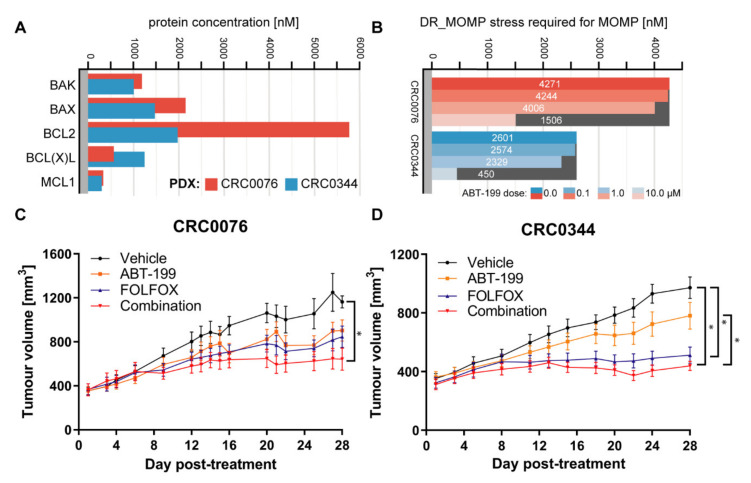
DR_MOMP successfully predicated the sensitivity of two CRC PDXs to treatment with FOLFOX and ABT-199 in combination. (**A**) Two mCRC PDXs (CRC0344 and CRC0076) were assessed for BCL-2 family proteins levels by quantitative Western blot (Appendix A). (**B**) The susceptibility of each model to undergo MOMP in response to genotoxic chemotherapy alone or in combination treatment with ABT-199 was calculated using DR_MOMP. (**C**,**D**) Anti-tumour activity of ABT-199 alone and in combination with FOLFOX chemotherapy in the two models was assessed in vivo. Mice bearing either a CRC0076 or CRC0344 tumour were treated with vehicle control, ABT-199 100 mg/kg, FOLFOX or the combination ABT-199 + FOLFOX; *n* = 13–14/group, error bars = s.e.m., * *p* ≤ 0.05. The effect of treatment on absolute tumour volumes of the predicted combination only responder CRC0076 and the predicted FOLFOX alone responder CRC0344 PDX models are shown respectively (**C**,**D**).

**Figure 3 cancers-12-02978-f003:**
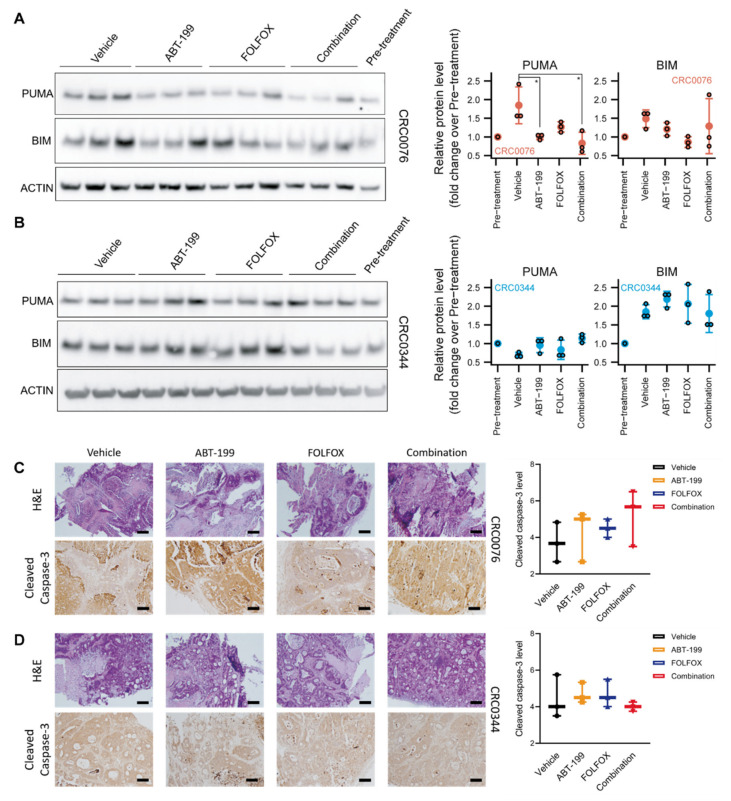
Decrease in p53 upregulated modulator of apoptosis (PUMA) levels but no effect on BCL2-Interacting Mediator of cell death (BIM) or cleaved caspase-3, following ABT-199 treatment in CRC0076 PDX model. (**A**,**B**) Western blot analysis comparing the levels of PUMA and BIM in CRC0076 (**A**) and CRC0344 (**B**) PDX models. β-actin or glyceraldehyde 3-phosphate dehydrogenase (GAPDH) (see Appendix A) were used as loading controls. Error bars represent mean +/− 2 SD, * *p* ≤ 0.05 (ANOVA, Tukey post-hoc). (**C**,**D**) Analysis of cleaved caspase-3 levels in CRC0076 (**C**) and CRC0344 (**D**) PDX models. Representative images of H&E (magnification 10×, scale bar 100 μm) and cleaved caspase-3 (magnification 5×, scale bar 200 μm) are shown for CRC0076 and CRC0344 in (**C**,**D**) respectively. The Allred method [15] was used to quantify the extent and intensity of staining, error bars represent mean +/− s.e.m (*n* = 3 tumours per treatment, *n* = 2–3 sections per tumour).

**Figure 4 cancers-12-02978-f004:**
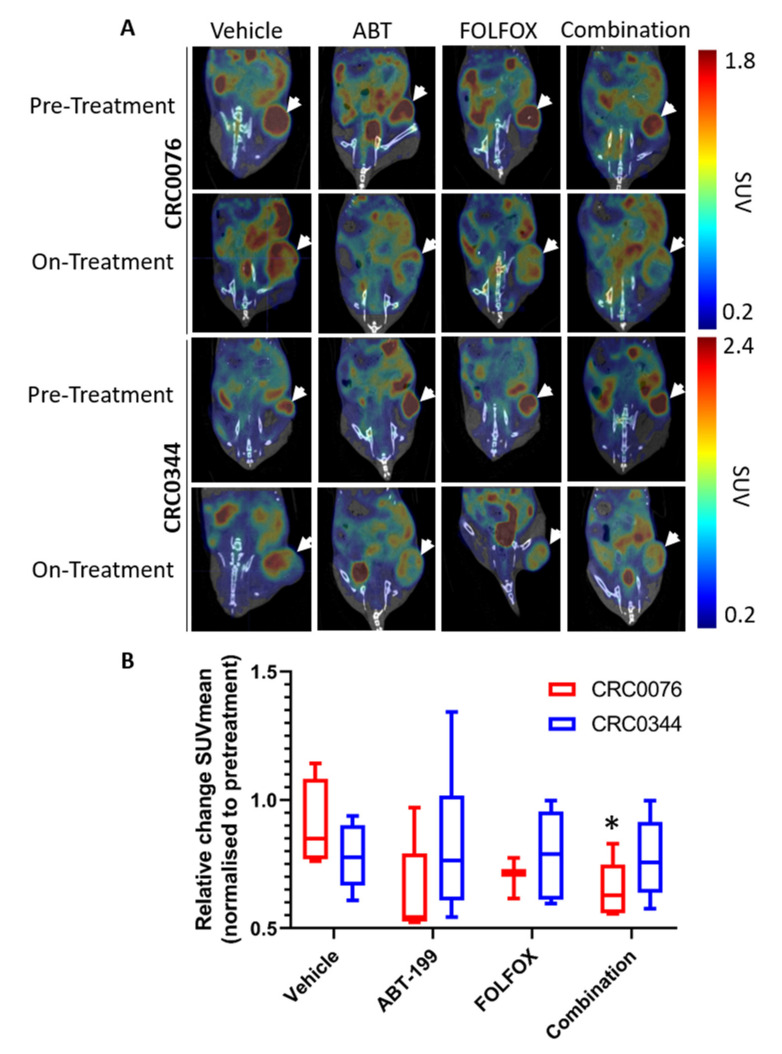
Analysis of glucose uptake and metabolism in the predicted combination responder CRC0076 and the predicted FOLFOX alone responder CRC0344 PDX models. For imaging, *n* = 6 mice/group/PDX (at pre-treatment time point) received ^18^F-FDG via tail vein injection, with image acquisition taking place 1 h after injection. Representative fused PET/CT images of CRC0076 and CRC0344 PDXs are shown (the same mice at pre- and on-treatment time points). White arrows indicate the tumours, scale bar indicates standard uptake values (SUVs) (**A**). N.B. Animals were imaged in a prone position. The coronal plane shown is positioned at the centre of the tumour which is located posterior to the bladder. Where the coronal plane intersects with the bladder, urinary excretion of FDG is observed as an intense signal. Comparison of relative change SUVmean at week 2 of treatment normalised to pre-treatment for CRC0076 and CRC0344 PDXs is presented. Error bars = s.e.m., * *p* ≤ 0.05 (paired *t*-tests, adjusted Holms) (**B**).

**Figure 5 cancers-12-02978-f005:**
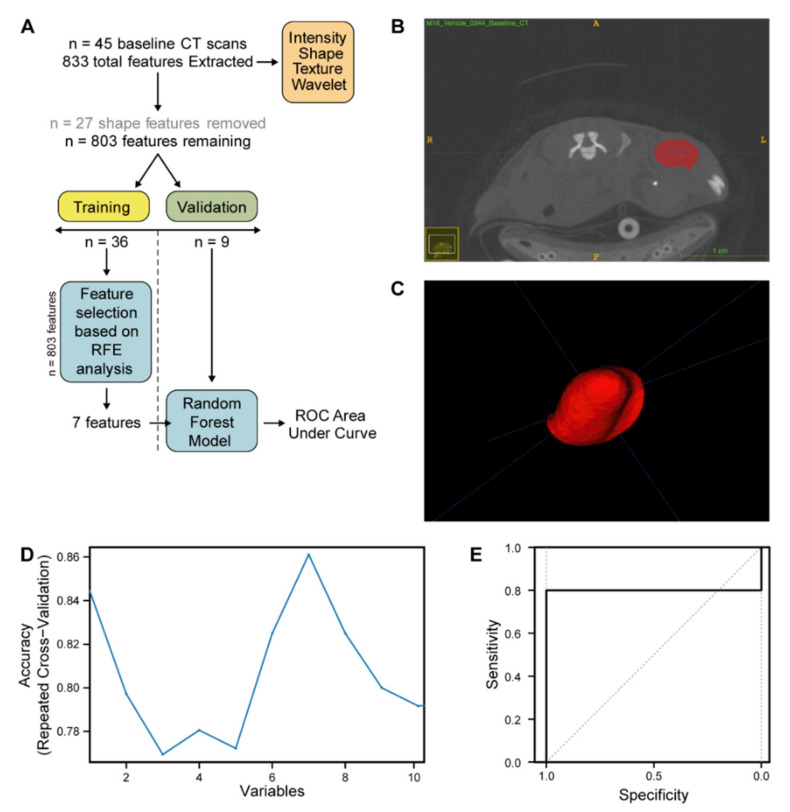
Radiomic analysis of pre-treatment CT images of CRC0076 and CRC0344 can successfully distinguish between the PDXs using seven unique radiographic features. (**A**) Visual Representation of PDX radiomic analysis workflow. *n* = 45 baseline CT scans were partitioned into a training dataset (*n* = 36) and validation dataset (*n* = 9) before Recursive Feature Elimination was performed on the training dataset. (**B**) Representative CT axial slice of a mouse implanted with CRC PDX CRC0344. Region of interest (ROI, in red) was drawn semi-automatically on every slice containing tumour to enable segmentations of the tumour for radiomic feature extraction. (**C**) 3D representation of the mask used to segment the tumour. (**D**) Output of the recursive feature elimination (RFE) method. Using an RFE algorithm the ideal number of features was identified as 7 for radiomic model construction. (**E**): Graph depicting the result of the receiver operating characteristic (ROC) area under the curve (AUC) analysis of the 7 features identified by the model in a test cohort (*n* = 9). AUC is 0.8 (95% Confidence interval 0.408–1).

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
