# Peer review of "Implementing Systems Modelling and Molecular Imaging to Predict the Efficacy of BCL-2 Inhibition in Colorectal Cancer Patient-Derived Xenograft Models"

_cancers, 2020, doi:10.3390/cancers12102978_

Round 1
Reviewer 1 Report
I do not have further comments.
Reviewer 2 Report
Authors seem to have addressed my comments, no further suggestions.
This manuscript is a resubmission of an earlier submission. The following is a list of the peer review reports and author responses from that submission.
Round 1
Reviewer 1 Report
In this manuscript O’Farrell et al. introduce DR_MOMP as a tool to predict CRC cells
sensitivity to undergo apoptosis. The analyzed the sensitivity HCT-116 and DLD-1 to the synergistic effect of ABT-199 and 5-FU/OX combination. Following, they analyzed the sensitivity of two PDX samples CRC0076 and CRC0344 to the same regimen. Then they applied noninvasive tools to predict the tumor sensitivity in vivo. I found this manuscript interesting and with translational potential. However, this manuscript complicated to understand as the flow was unclear, and thus will require substantial editing.
Major comments
- In figure S1B, the DR_MOMP calculated “a similar apoptosis sensitivity for both cell lines” (row 107). The meaning of two cell lines that demonstrate the same sensitivity is unclear. This experiment needs to be conducted on CRC cell lines that are predicted by DR_MOMP not to have the same apoptosis sensitivity. In addition, two cell lines are not sufficient; at least three from each group is needed.
- Figure S1C is confusing. This figure needs to be presented as EC50 plots and not as a bar graph.
- The authors demonstrated that both PDXs behaved differently in their response to FOLFOX treatment (Figure 3A and B). Then they demonstrated a difference in PUMA and BIN expression upon the different treatments between the two PDXs. However, they failed to show differences in cleaved caspases levels (Figure 3C and D). What do the authors speculate regarding the cellular mechanism that induces apoptosis upon the different treatments? In addition, both PUMA and BIN are proapoptotic factors, if so, why is their expression is reduced in CRC0076, which is more sensitive to ABT-199? This needs to be explained.
Minor comments
- Figure 1 is a schematic representation of the experimental workflow. The right side of the scheme is unclear and needs to be simplified.
- Could the authors explain the inconsistency between figure S1B and Table S2?
- In row 105, they wrote,” DR_MOMP calculated a similar apoptosis sensitivity of HCT-116 and DLD-1 cells. This stamen needs to be elaborate, how do they define “similar apoptosis sensitivity”, is this a statistic term? Because in figure S1B, there is a difference between DLD1 and HCT116.
- In figure S1, there is an inconsistency between the panel numbers in the figure to their name in the legend.
- Row 113 lacks any reference to a figure.
- Row 152 it should be figure 3B and not 3C
- According to the manuscript flow figures, 3C and D needs to be presented after figure 4.
- The colors representing the different PDXs are inconsistent between Figures 2 and 3D.
- In figure 2A the authors demonstrate a quantified level of selected proteins between CRC0076 and CRC0344. They need to show, at least as a supplement figure, a representative WB image of these proteins in the different PDX.
- In figure S1C in both DLD-1 and HCT-116 samples, the combined treatment of 0.2µM Ox plus 1µM 5-FU, and ABT-199 protected the cells from only ABT-199 treatment, do the authors have an explanation for this?
- In row 76, they write CRC but explain its acronyms only on row 77.
- In row 156, is the reference to Figure 3D correct?
- Row 140, CDC0076 are not highly resistant (unproven) but more ×´resistant than×´
- Please, report the full-length name of all acronyms used, especially genes and methods; some of them are missing (for example, ROC).
- Fix row 392
Reviewer 2 Report
- Figure 1: “predictive biomarker for ABT-199”, please rephrase, as I understand it is not a biomarker for ABT-199 but a prediction for ABT-199 effectiveness
- Line 144ff. It would be interesting to know what engraftment rate is typical in this model and whether the software outcome could somehow predict this (e.g. were there lines that were predicted to be very sensitive but didn’t engraft well?)
- Figure 3C, unclear, pre-treatment images are needed for each condition, even if that goes in supplemental, it is hard to see any effect if there is only one representative pre-treatment image and no direct comparison possible.
- Line 178 ff, although well-known PUMA and BIM should be spelled out
- Line 187 “this is not seen in the CRC0344 model (Figure 4D)” and sentence before that. This sentence is not supported by the presented data (images or graph). Insets are required for Figure 4C/D. No increase in expression can be seen in the images that were chosen and the variance of data in the combination treatment is too high (graph) to speculate on this.
- Line 200ff “..were performed in the same animals (Figure 3C).” This is not true, Figure 3C does not show this as pre-treatment images are not shown for each condition , they are need in order todraw any conclusion (see also above)
- Discussion lacks explanation about the suggested (although not visible) increase in cleaved caspase 3 expression for CRC0076 but not CRC0344
- Spell check required: e.g. tumor vs tumour e.g. line 317 and 319 but can be found in the whole document, a mix of AE and BE which is rather annoying.
- Line 354 “Zinn et al” needs dot after al
- Figure 1: “predictive biomarker for ABT-199”, please rephrase, as I understand it is not a biomarker for ABT-199 but a prediction for ABT-199 effectiveness
- Line 144ff. It would be interesting to know what engraftment rate is typical in this model and whether the software outcome could somehow predict this (e.g. were there lines that were predicted to be very sensitive but didn’t engraft well?)
- Figure 3C, unclear, pretreament images are needed for each condition, even if that goes in supplemental, it is hard to see any effect if there is only one representative pre-treatment image and no direct comparison possible.
- Line 178 ff, although well-known PUMA and BIM should be spelled out
- Line 187 “this is not seen in the CRC0344 model (Figure 4D)” and sentence before that. This sentence is not supported by the presented data (images or graph). Insets are required for Figure 4C/D. No increase in expression can be seen in the images that were chosen and the variance of data in the combination treatment is too high (graph) to speculate on this.
- Line 200ff “..were performed in the same animals (Figure 3C).” This is not true, Figure 3C does not show this as pre-treatment images are not shown for each condition , they are need in order todraw any conclusion (see also above)
- Discussion lacks explanation about the suggested (although not visible) increase in cleaved caspase 3 expression for CRC0076 but not CRC0344
- Spell check required: e.g. tumor vs tumour e.g. line 317 and 319 but can be found in the whole document, a mix of AE and BE which is rather annoying.
- Line 354 “Zinn et al” needs dot after al
- Can the data be compared to patient dat? E.g. if software predicted outcome for the two cell lines, can the patients treatment or disease outcome say anything about the obtained data?
- Are there any CT scans of patients (e.g. glucose uptake) and correlated outcome data that can inform about the relevance of the finding s in the PDX model?
- Line 392 and line 501: ex vivo is crossed?
- Can the data be compared to patient dat? E.g. if software predicted outcome for the two cell lines, can the patients treatment or disease outcome say anything about the obtained data?
- Are there any CT scans of patients (e.g. glucose uptake) and correlated outcome data that can inform about the relevance of the finding s in the PDX model?
- Line 392 and line 501: ex vivo is crossed?
Reviewer 3 Report
The authors describe effectiveness of BCL-2 antagonist, ABT-199, against colorectal cancer tumor models. They also presented that in vitro experimental prediction by DR-MOMP reflects in vivo results. Moreover, they showed FDG-PET analysis performed at 2-weeks treatment could predict the final situation of ABT-199 treatment. The reviewer agree that in vivo models to predict the drug response in a real clinical situation is mandatory for the era of precision medicine. However, only two PDX models could not make the conclusions that the authors presented and described as in their limitation. The manuscript is immature to present as a research article at this form.
- The reviewer cannot understand the result of Fig4C. The IHC level of cleaved caspase 3 seems to be lower in combination arm rather than ABT199 alone.
- The reviewer cannot understand the result of Fig3C. Why FDG-PET uptake decreased in vehicle arm in comparison to pre-treatment? At least the vehicle arm should be the same or increase in the uptake.
- The reviewer wants to know the expression of Ki67 in all the arms presented. The authors just presented proteins related to apoptosis. Is there any possibility that BIM or PUMA do not participate in reduction of the tumor growth by ABT-199 treatment?
- BCL2 expression levels between CRC cell lines (DLD-1 and HCT-116) and the PDX cells (CRC0076 and CRC0344) were quite different. Do the authors think that the experiment presented in their cell lines can apply to PDX models? The level of DR-MOMP stress required for MOMP is also quite different.